# Fabrication of Modified Polyurethane Sponge with Excellent Flame Retardant and the Modification Mechanism

Hang Li [1,†], Chen-Yang Zhang [1,2,†], Ya-Ling Yu [2], Chang-Jin Liang [2], Guang-Ming Yuan [3], Huan Yang [2], Yun-Ying Wu [2] and Shao-Min Lin [1,2,*]

1   Chaozhou Branch of Chemistry and Chemical Engineering Guangdong Laboratory, Chaozhou 521041, China
2   School of Materials Science and Engineering, Hanshan Normal University, Chaozhou 521041, China
3   School of Materials Science and Engineering, Central South University of Forestry and Technology, Changsha 410018, China
*   Correspondence: lsm678@hstc.edu.cn
†   These authors contributed equally to this work.

**Abstract:** Research on polyurethane sponge (PUS), a widely used polymer material, and its flame-retardant performance is of great significance. In this study, PUS was modified to prepare a highly efficient flame-retardant composite using a soaking method. The PUS nearly vanished at 11 s after ignition, and the solid residue rate of the PUS was 5.65 wt% at 750 °C. The net structure, composed of nano $SiO_2$, was maintained in the modified PUS at 750 °C, and the solid residue rate was 69.23%. The maximum HRR of the PUS decreased from 617 W/g to 40 W/g and the THR of the sample reduced from 33 kJ/g to 9 kJ/g after modification. The results suggested that the modified PUS gained excellent flame-retardant performance. The flame-retardant layer in the modified PUS was amorphous. The surface of the modified PUS was rich in Si, O, and C elements and lacked a N element, suggesting that inorganic flame retardants were abundant on the surface layer of the modified PUS. The Si-O-C vibration and Si-O-Si stretching in the modified PUS indicates that the organic–inorganic hybrid structure formed on the PUS surface, which could be attributed to the polymerization and condensation of the silica precursor. Thus, the modified PUS provided an excellent flame-retardant layer. The results are of interest for producing efficient flame-retardant PUS using a simple method.

**Keywords:** layer-by-layer assembly; sol-gel methods; graphene oxide; silica; flame retardant; polyurethane sponge

## 1. Introduction

Polyurethane sponge (PUS) is a loose porous polymer material with high porosity, low density, and high elasticity. PUS is usually used to prepare excellent functional materials, such as superhydrophobic sponge dressings [1,2], solar absorption materials [3], and oil/water separation materials [4–6]. Therefore, PUS is widely applied in the construction, motor, aviation, and ceramics industries [7].

PUS is prone to decomposition and burning when it approaches a high-temperature heat source [8,9]. During combustion, the melting drop phenomenon occurs, and large amounts of toxic and harmful gases are released, so PUS does not easily meet the needs of industrial applications [10]. The general way to improve the flame-retardant performance of PUS is to introduce flame-retardant agents, such as chlorine, bromine, and a series of phosphorus flame-retardant agents during the foaming process of the sponge. Jiang et al. prepared an alginate hybrid sponge material with flame-retardant properties using a simple chemical foaming method in situ [11]. However, the combustion of the products released a large amount of acids and strong corrosive gases, which are harmful to the environment and the human body [12].

The surface modification of PUS products has received much attention for improving flame-retardant performance. Liu et al. used a polydopamine polymerization-coated

polyurethane sponge and then treated it with hexamethyldisilazane (HMDS) to prepare polyurethane sponge with excellent flame-retardant performance [13]. J. Miao et al. fabricated an interpenetrating polymer network between PUS and porous organic polymers containing P and N elements, which enhanced the flame-retardant performance and mechanical strength of the samples [14]. Jamsaz et al. prepared a flame-retardant graphene-based polyurethane sponge by functionalizing it with graphene oxide (GO) [15]. The organic modification of PUS can improve flame-retardant performance, but a lot of smoke during combustion is produced. It also has poor flame-retardant properties. Thus, inorganic flame-retardant agents, such as magnesium hydroxide [16], titanium dioxide [17,18], and silicon dioxide [19–22], have been investigated by a few researchers. Wang et al. prepared rigid polyurethane foams containing SiO2 nanospheres/graphene oxide hybrids and dimethyl methylphosphonate using a free-form foaming method, and the flame-retardant properties of the materials were improved with a 44.6% reduction in the pHRR [23]. In addition, the production of hydrogen chloride acid gas was inhibited, a small amount of toxic and harmful gases was produced, and the thermal stability was good at high temperatures. However, the process was complicated and costly.

In this study, the PUS was modified with organic materials (graphene oxide and polystyrene sulfonate) and an inorganic material ($SiO_2$) via a soaking method. The stable structure and the inorganic layer consisting of $SiO_2$ were expected to form on the surface of the PUS. The thermal behavior and microstructure of the PUS-based composites were investigated to explore the flame-retardant performance and the modification mechanism.

## 2. Materials and Methods

### 2.1. Materials Preparation

The PUS (Yongjia Sponge Products Company, Changzhou, China) was modified by simple soaking methods. A schematic diagram of the preparation process of the PUS-based composite is shown in Figure 1. The preparation steps were as follows:

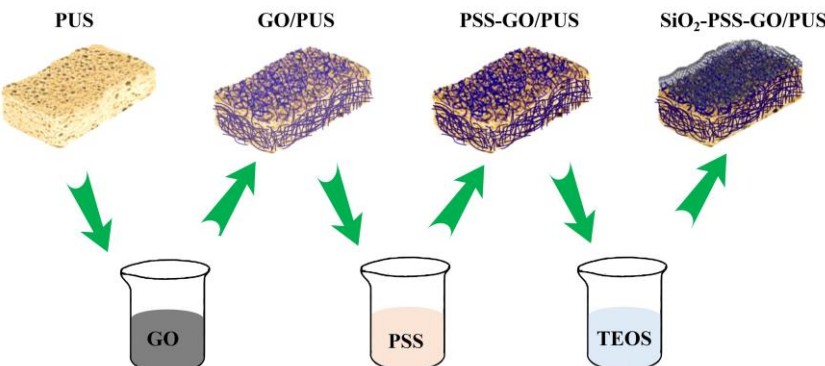

**Figure 1.** Schematic diagram of the preparation process of modified PUS.

Step 1: The PUS was cleaned with deionized water and ethanol (99%, Fuchen Chemical Reagents Factory, Tianjin, China) to remove the impurities on the surface of the material.

Step 2: The PUS was soaked in 2 wt% graphene oxide (>99%, Shanghai Aladdin Biochemical Technology Co., Ltd., Shanghai, China) aqueous solution under a vacuum (0.08 MPa) for 2 h at 40 °C. The PUS showed a large amount of positive charge. Then, the treated sample was soaked in 2 wt% polystyrene sulfonate (PSS, average Mw ~70,000, Shanghai Aladdin Biochemical Technology Co., Ltd., Shanghai, China) aqueous solution under a vacuum (0.08 MPa) for 2 h at 40 °C. The graphene oxide was stable on the PUS surface through electrostatic adsorption.

Step 3: A mixture of tetraethyl orthosilicate (TEOS, >99%, Xilong Scientific, China) and deionized water with a mass ratio of 1:1 was stirred for 30 min at 30 °C to obtain the silica precursor solution; during the mixing process, diluted hydrochloric acid (HCl, >99%, Xilong Scientific, Shantou, China) was slowly added to make the solution

pH = 2. By adjusting the ratio of TEOS and water and the solution's pH and temperature, the silica particle size of the silica formed by hydrolysis and the polycondensation could be controlled.

Step 4: GO/PUS was soaked in the silica precursor solution for 12 h under a vacuum (0.08 MPa) at 35 °C; finally, the modified material was rinsed with deionized water and dried in the oven at 50 °C to obtain the modified PUS.

### 2.2. Methods

The heat release rate (HRR), total heat release (THR), and heat release capacity (HRC) were measured using a microcalorimeter (FAA-PCFC, Fire Testing Technology, Britain) with a heating rate of 1 °C/s between 100 °C and 750 °C. A cone calorimeter (CONE, FTT0007, Fire Testing Technology, West Sussex, UK) was used to measure the fire behavior of the samples in accordance with IOS 5660. To eliminate sources of error, each specimen was tested three times.

The thermal behavior of the samples was investigated using thermal gravity analysis (TGA) in the air with a thermal analyzer (STA 449 F3, NETZSCH, Bavaria, Germany) with a heating rate of 10 °C/min between 25 °C and 750 °C.

The microstructure of the samples was examined using a SU 5000 scanning electron microscope (Hitachi, Tokyo, Japan) with a field-emission gun that normally operated at a 5–10 kV acceleration voltage in a high-vacuum environment. Energy-dispersive spectroscopy (EDS; Quantax, Bruker, Germany) was employed to study the distribution of elements along the surface of the samples.

Fourier transform infrared spectroscopy (FTIR) was conducted on an INVENIO-S spectrometer (Bruker, German) in the MIR range (4000–400 cm$^{-1}$). X-ray powder diffraction (XRD) patterns were obtained using a Miniflex600 diffractometer (RIGAKU, Tokyo, Japan) that operated at 30 kV and 10 mA with Cu Kα (0.15418 nm) filtered radiation, a curved graphite secondary monochromator, a scan range of 10°2θ to 60°2θ, a step width of 0.02°2θ, and a scan speed of 10°/min.

### 3. Results and Discussion

#### 3.1. The Flame-Retardant Performance of the Modified PUS

The combustion performance of the PUS before and after modification is shown in Figure 2. A flame was observed at 1 s after ignition for the PUS. The melts dropped from the PUS at 3 s, and the PUS burned completely at 11 s (Figure 2a). In the modified PUS, the tip of the sample burned red in the center of the flame at 2 s, and there was no obvious flame or melts ((Figure 2b)). The shape of the modified PUS was intact in the fire until the end of the experiment.

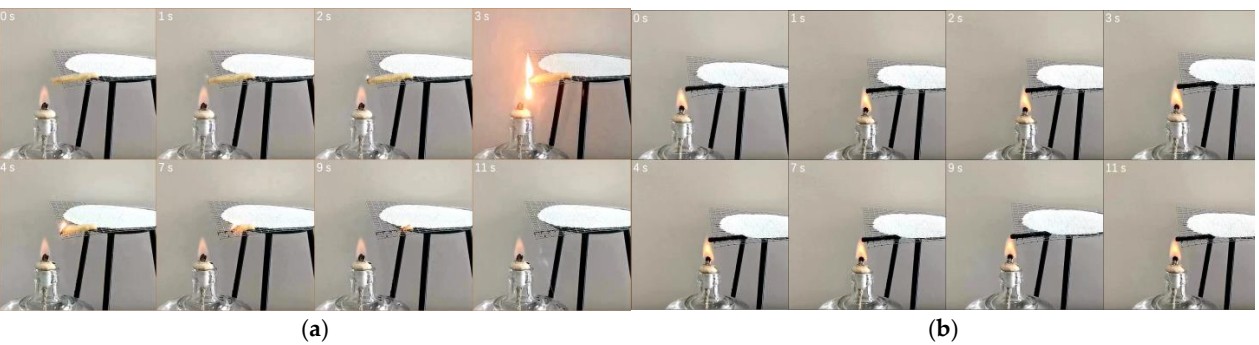

(**a**)                                                                            (**b**)

**Figure 2.** The pictures of (**a**) PUS and (**b**) modified PUS burned in the air at different times.

The flame-retardant performance of the PUS was macroscopically observed in the alcohol lamp combustion experiment. The PUS provided an excellent flame retardant after modification. Molten drops were not produced, and the modified PUS had good

smoke-suppression performance. At the same time, pollution in the air was reduced during the process of combustion.

The flammability of the samples was evaluated using cone calorimetry. Some parameters used in this test include smoke produce rate (SPR), total smoke release (TSR), average CO yield (CO-Y), average $CO_2$ yield ($CO_2$-Y), time to ignition (TTI), and ignition to flameout (ITF). Compared with the PUS, the modified PUS had longer TTI and ITF (Table 1) values because the modified PUS exhibited sustained burning with weak flames. The main function of flame retardants is to suppress the production of toxic smoke, providing time for people to safely evacuate the premises. The modification of the PUS reduced the maximum SPR of the PUS from 0.0595 $m^2/m^2$ to 0.0169 $m^2/m^2$, and the modification of the PUS significantly reduced the TSR of the PUS from 213 $m^2/m^2$ to 51 $m^2/m^2$ (Figure 3). Both peaks of CO-Y and $CO_2$-Y significantly decreased after modification (Table 1). This implied that the modified surface served as an "insulating blanket" to intercept the exchangeable channel of heat flux and oxygen.

**Table 1.** Data from cone tests.

| Samples | TTI (s) | ITF (s) | CO-Y (kg/kg) | $CO_2$-Y (kg/kg) |
|---|---|---|---|---|
| PUS | 2 | 98 | 2.56 | 222.83 |
| Modified PUS | 29 | 154 | 0.16 | 1.51 |

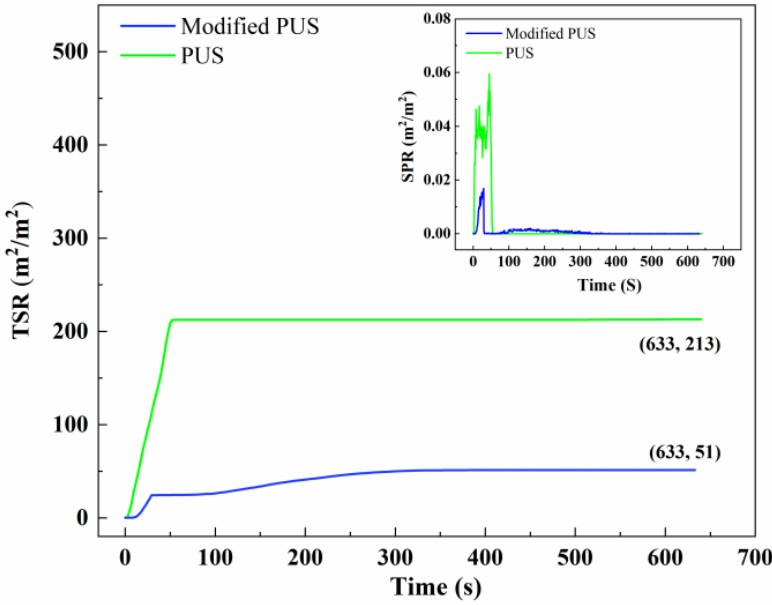

**Figure 3.** The total smoke release (TSR) and the smoke produce rate (SPR) curves of PUS before and after modification.

The micromorphology of the PUS and modified PUS at 750 °C is shown in Figure 4. The amount of the carbon residue of the PUS was tiny (Figure 4b) and distorted (Figure 4e,f) at 750 °C. The net structure of the modified PUS was maintained at 750 °C (Figure 4d). The nanoparticle layer of the modified PUS existed after calcination at 750 °C (Figure 4g,h). The microstructure of the modified PUS was shown in the Figure 5a; the corresponding element mapping of the modified PUS at 750 °C indicated the residue was composed of Si and O (Figure 5b,c and Table 2). The XRD pattern of the residue suggested that the residue was amorphous (Figure 5d). The $SiO_2$ layer on the surface prevented the rapid oxidation of the organic materials during combustion (Figure 2b).

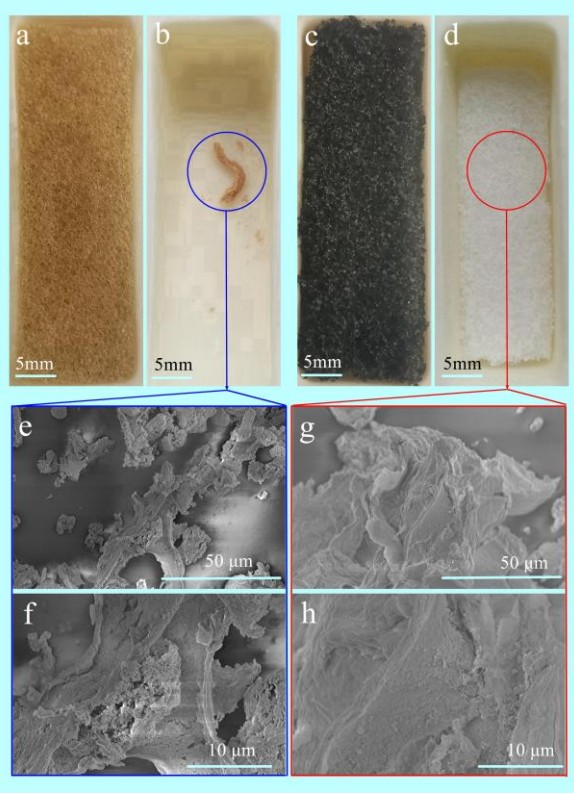

**Figure 4.** The image of PUS (**a**,**b**,**e**,**f**) before and (**c**,**d**,**g**,**h**) after modification at different temperatures (**a**,**c**) 20 °C; (**b**,**d**,**e**,**f**,**g**,**h**) 750 °C.

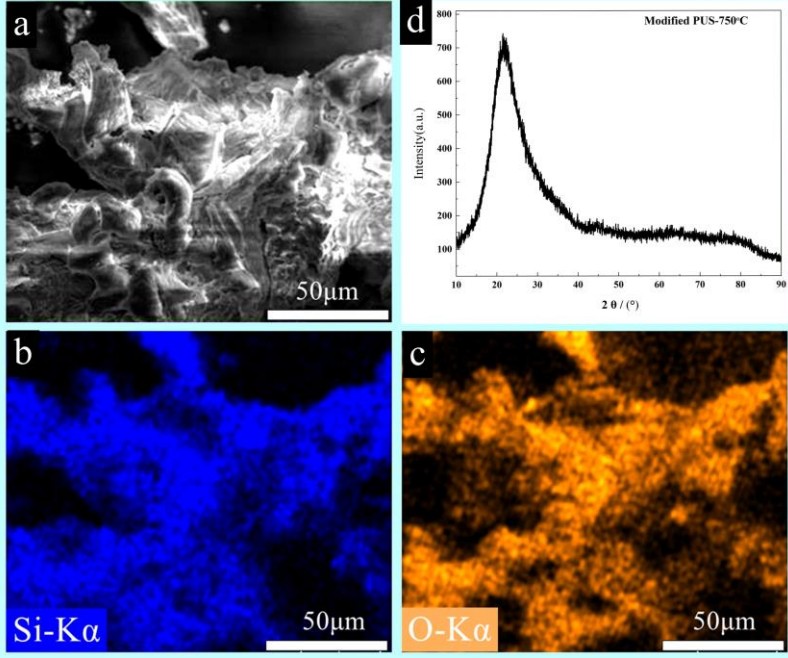

**Figure 5.** (**a**–**c**) The element mapping and (**d**) XRD pattern of the modified PUS at 750 °C.

**Table 2.** The elemental composition of modified PUS at 750 °C.

| Element | Ratio of Atom/% |
|---------|-----------------|
| O | 67.36 |
| Si | 32.64 |

The results of the thermal gravity analysis of the sample are shown in Figure 6. The decomposition of the sample was divided into four stages in the nitrogen atmosphere. At 40~240 °C, the weight of the PUS changed little, and the mass loss of the modified PUS changed a little (Figure 6a), probably due to the loss of water adsorbed by the $SiO_2$. At 240~316 °C, the mass loss of the PUS sharply increased with the temperature, and the maximum mass loss rate ($DTG_{max}$) reached 11.8 wt%/min at 306 °C (Table 3); the mass loss of the modified PUS slowly increased with increasing temperature, and the maximum mass loss rate ($DTG_{max}$) reached 0.9 wt%/min at 298 °C (Figure 6b). At 316~400 °C, the mass loss rate of the PUS and the modified PUS increased first and then reduced; the maximum mass loss rate of the sample reduced from 9.4 wt%/min at 384 °C to 1.6 wt%/min at 354 °C after modification due to the delayed transfer of heat and mass in the modified PUS with inorganic silica. At 400~750 °C, the mass of the sample slightly changed (Figures 5b and 6a); the solid residue rate of the PUS and modified PUS was 5.65 wt% and 69.23% at 750 °C, respectively. This indicated that the modified PUS experienced a lower mass loss and had high thermal stability.

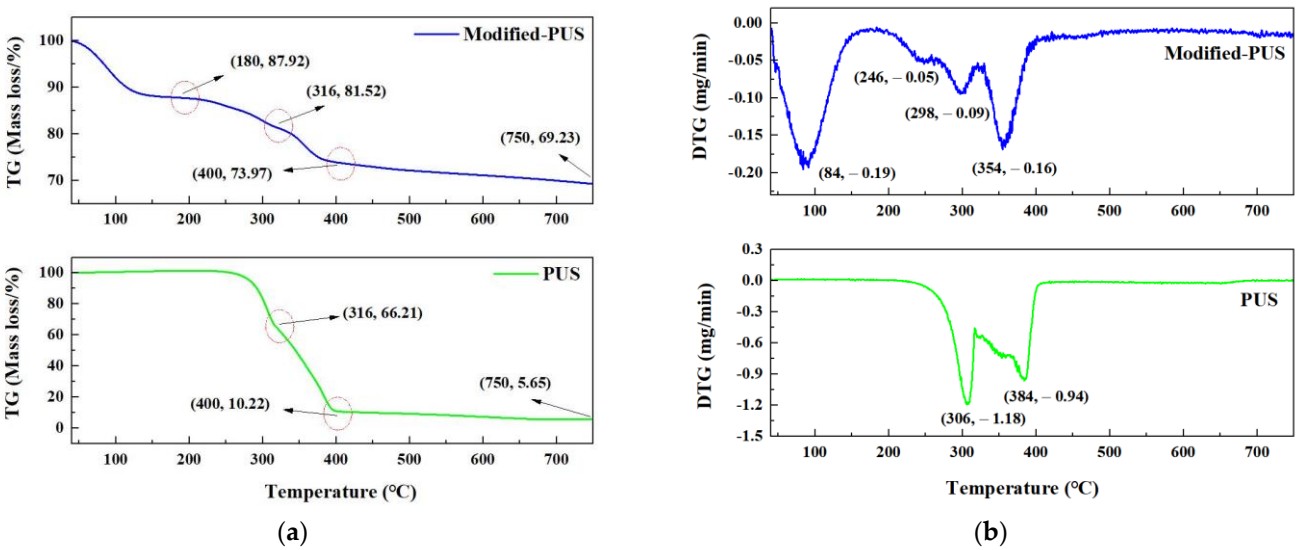

**Figure 6.** (**a**) The thermal gravity (TG) and (**b**) differential thermal gravity (DTG) curves of PUS before and after modified PUS.

**Table 3.** The thermal weight data of the sample.

| Parameter | PUS | Modified PUS |
|-----------|-----|--------------|
| Solid residue rate (wt%) | 5.65 | 57.29 |
| $DTG_{max}$ (wt%/min) | 1.18 | 0.16 |
| $T_{max}$ (°C) | 306 | 354 |

The heat release rate (HRR) and the total heat release (THR) are shown in Figure 7 and Table 4. The HRR peaks of the PUS were 150 W/g and 617 W/g. The HRR peaks of the modified PUS were 21 W/g and 40 W/g, which were 86% and 94% lower than that of the PUS, respectively. The THR of the sample reduced from 33 kJ/g to 9 kJ/g after modification. The result suggested that the modified PUS provided excellent flame-retardant performance, probably due to the introduction of $SiO_2$ on the PUS.

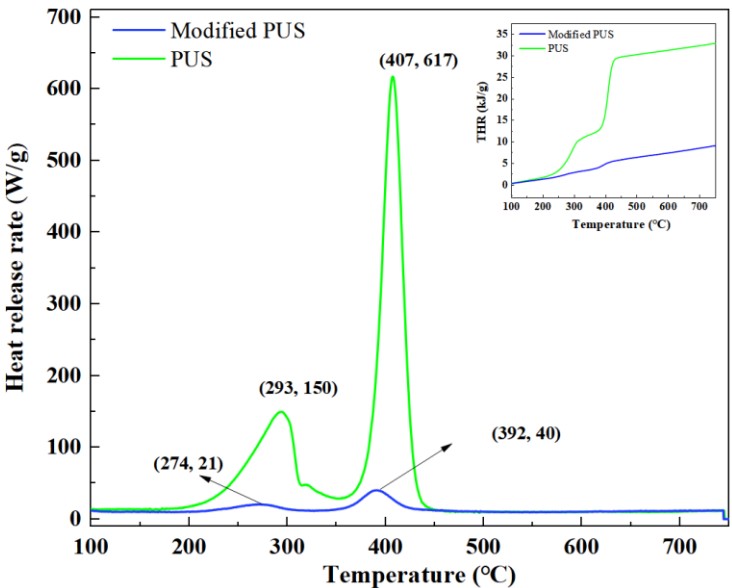

**Figure 7.** The heat release rate (HRR) and total heat release (THR) curves of PUS before and after modification.

**Table 4.** The total heat release (THR) and the peaks of heat release rate (HRR) curve of PUS before and after modification.

| Samples | THR(kJ/g) | HRR (W/g) | | Temperature (°C) | |
|---|---|---|---|---|---|
| | | $P_1$ | $P_2$ | $P_1$ | $P_2$ |
| PUS | 33 | 150 | 617 | 293 | 407 |
| Modified PUS | 9 | 21 | 40 | 274 | 392 |

### 3.2. The Modification Mechanism of the PUS

A crystal diffraction peak was not observed in the PUS or the modified PUS (Figure 8). Compared with that of the PUS, the peak of the modified PUS shifted left, indicating that the formation of the flame-retardant layer on the PUS changed the structure of the material [24]. The result suggested that the flame-retardant layer in the modified PUS was amorphous.

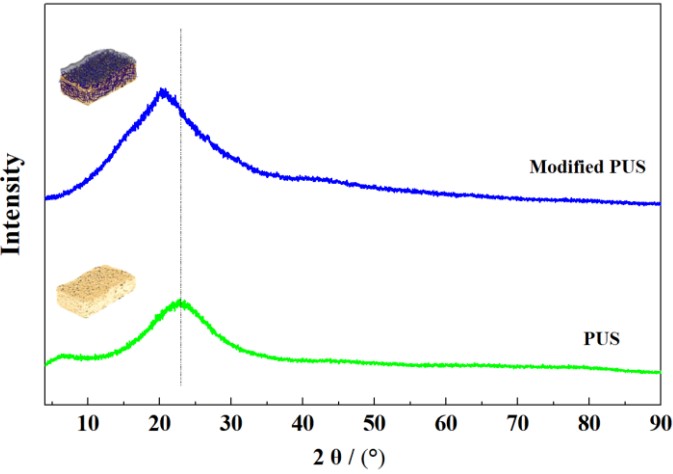

**Figure 8.** The XRD patterns of the PUS before and after modification.

The morphology of the sample was investigated using SEM (Figure 9). The PUS had a rich three-dimensional porous network structure (Figure 9a), and the surface of the PUS was smooth (Figure 9b). The 3D network structure was maintained in the modified PUS (Figure 9c), while the modified PUS surface was coarse, and a few cracks were observed on the surface (Figure 9d). The cracks could have resulted from the inhomogeneous polymerization of the silica precursor. The C, O, and N elements were detected in the element mapping of the PUS (Figure 10a), while Si was observed in the modified PUS. The elemental composition (the ratio of atom, percentage) of the surface of the PUS and modified PUS is shown in Table 5. Compared with that of the PUS, the surface of the modified PUS was rich in Si and lacked N, suggesting that inorganic flame retardants were abundant on the surface layer of the modified PUS.

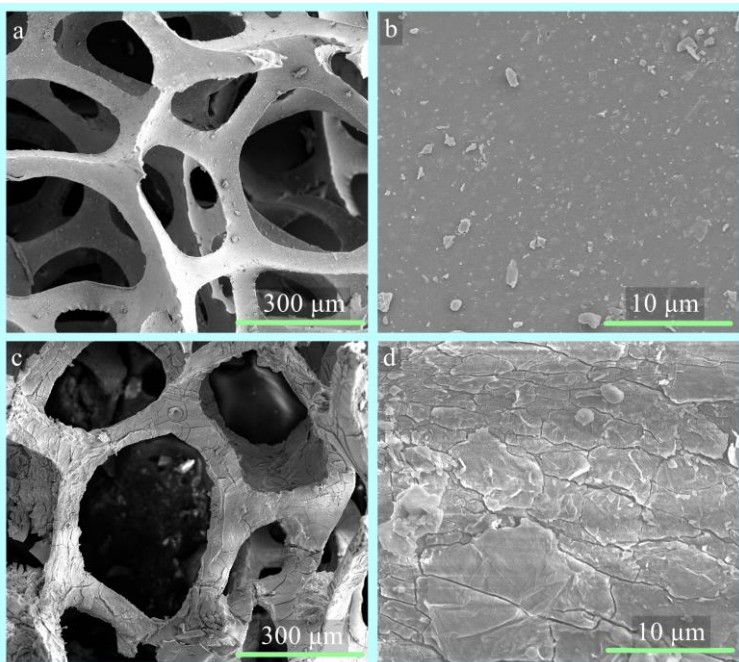

**Figure 9.** The SEM images of the (**a**,**b**) PUS and (**c**,**d**) modified PUS.

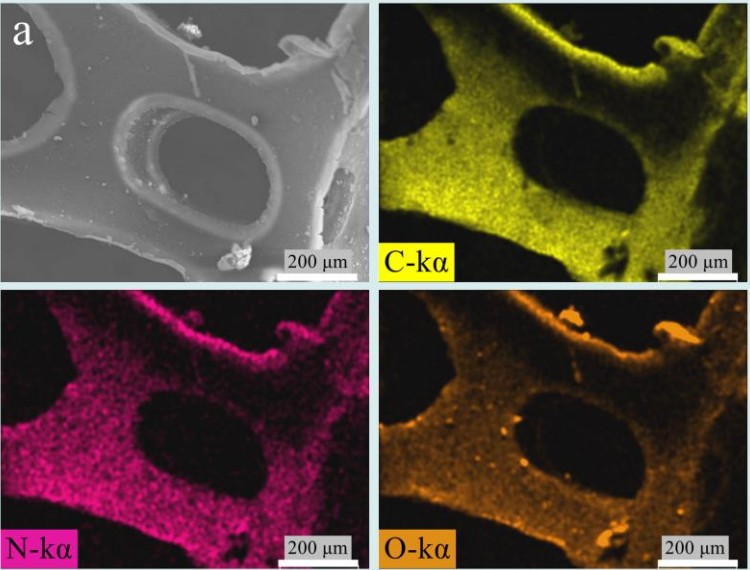

**Figure 10.** *Cont.*

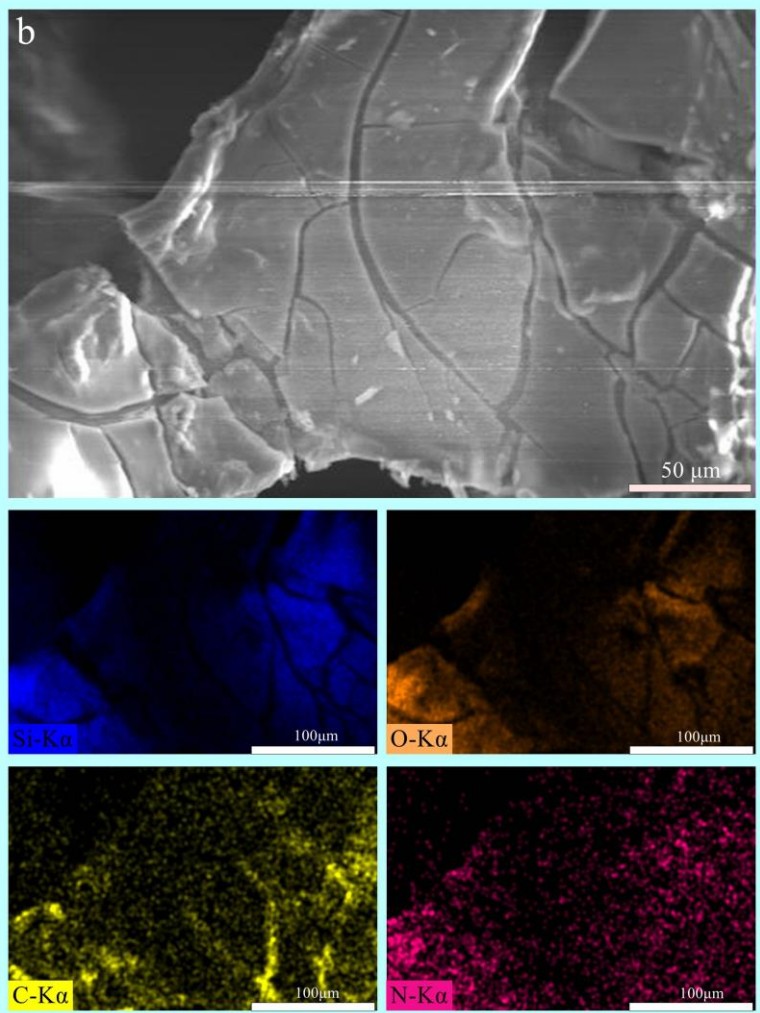

**Figure 10.** Element mapping of the (**a**) PUS and (**b**) modified PUS.

**Table 5.** The elemental composition (the ratio of atom, %) of PUS and modified PUS.

| Element | PUS | Modified PUS |
|---|---|---|
| C | 55.91 | 29.01 |
| N | 19.05 | 4.68 |
| O | 29.92 | 46.48 |
| Si | – | 19.84 |

The FTIR spectra of the PUS before and after modification are shown in Figure 11. The bands at approximately $3462$ cm$^{-1}$ and $1640$ cm$^{-1}$ were attributed to the stretching vibration and bending vibration of O-H. After modification, Si-O-C vibration ($\sim 1180$ cm$^{-1}$), Si-CH$_3$ vibration ($\sim 799$ cm$^{-1}$), Si-OH vibration ($\sim 949$ cm$^{-1}$), Si-O-Si stretching ($\sim 1084$ cm$^{-1}$), and Si-O-Si deformation ($\sim 460$ cm$^{-1}$) were found in the spectra. The result implied that SiO$_2$ had formed in the modified PUS. The band near $1170$ cm$^{-1}$ and the overlap of the band at $\sim 1170$ cm$^{-1}$ and $\sim 1082$ cm$^{-1}$ implied that the organic materials and the inorganic materials formed an organic–inorganic hybrid structure on the PUS surface [25–29].

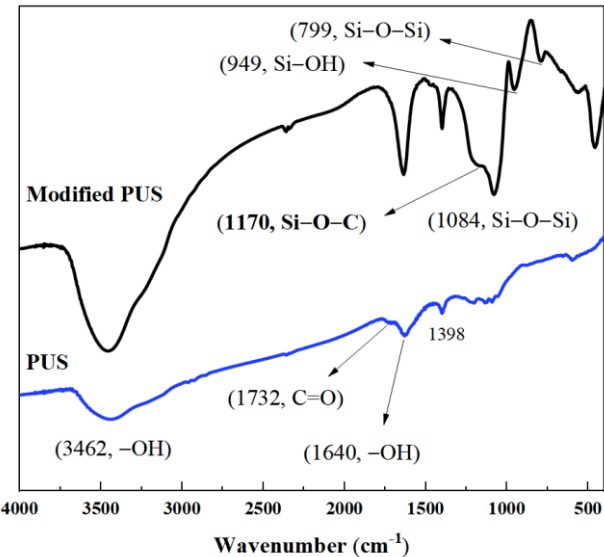

**Figure 11.** The FTIR spectra of PUS before and after modification.

The results for the PUS and modified PUS indicated that the PUS was mainly modified by $SiO_2$. The surface of the PUS was activated by GO and PSS, and the silica precursor was absorbed into the PUS (Figure 12). Part of the silica precursor was directly polymerized on the PUS, and part of the silica precursor condensed after chemical adsorption. The inconsistent behavior of the silica precursor contributed to the cracks on the surface of the PUS. The polymerization and condensation of the silica precursor resulted in the formation of an excellent inorganic ($SiO_2$) flame-retardant layer.

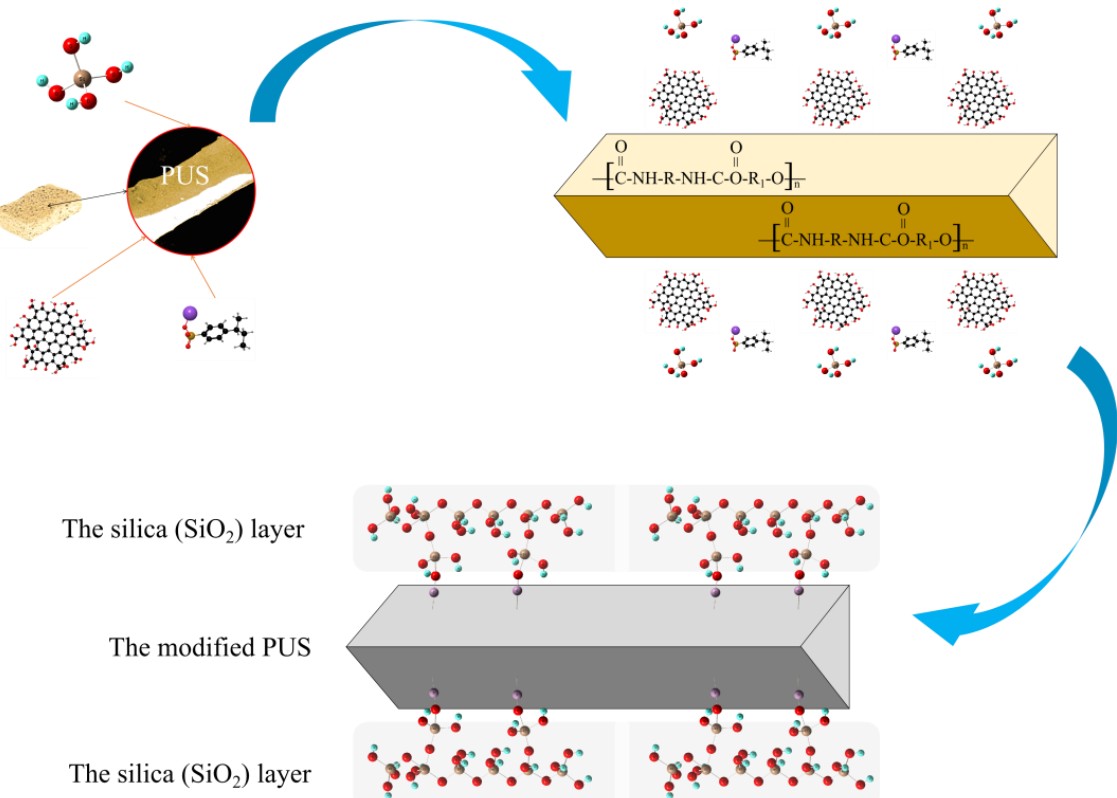

**Figure 12.** The schematic illustration of the modification mechanism of PUS with TEOS.

## 4. Conclusions

The efficient flame-retardant composite was prepared by modifying PUS with $SiO_2$. Fourier transform infrared spectroscopy (FTIR), X-ray powder diffraction (XRD), scanning electron microscopy (SEM), energy dispersive spectroscopy (EDS), thermogravimetric analysis (TGA), and microscale combustion calorimetry (MCC) were used to study the sample before and after modification. The shape of the modified PUS was intact in the fire, while the PUS was completely burned at 11 sec after ignition. The amount of carbon residue of the PUS was tiny, and the solid residue rate of the PUS was 5.65 wt% at 750 °C. The net structure, consisting of nano $SiO_2$, was maintained in the modified PUS at 750 °C, and the solid residue rate of the modified PUS was 69.23%. The maximum HRR of the PUS decreased from 617 W/g to 40 W/g, and the THR of the sample reduced from 33 kJ/g to 9 kJ/g after modification. The result suggested that the modified PUS provided excellent flame-retardant performance. The flame-retardant layer in the modified PUS was amorphous. The surface of the modified PUS was rich in Si, O, and C elements and lacked N, suggesting that inorganic flame retardants were abundant on the surface layer of the modified PUS. Si-O-C vibration (~1180 $cm^{-1}$) and Si-O-Si stretching (~1084 $cm^{-1}$) were found in the FTIR spectra of the modified PUS. The result indicated the $SiO_2$ formed in the modified PUS, and an organic–inorganic hybrid structure formed on the PUS surface, which was attributed to the polymerization and condensation of the silica precursor. Thus, the modified PUS provided an excellent flame-retardant layer. The modified PUS was prepared using the simple technology of soaking in a prepared solution, which is of interest in producing highly efficient flame-retardant materials with a low-cost method.

**Author Contributions:** Conceptualization, H.L. and C.-Y.Z.; investigation, Y.-L.Y.; methodology, C.-J.L.; validation, G.-M.Y.; formal analysis, H.Y.; data curation, H.Y. and Y.-Y.W.; writing—original draft preparation, H.L. and C.-Y.Z.; writing—review and editing, S.-M.L. All authors have read and agreed to the published version of the manuscript.

**Funding:** This research was supported by Chaozhou Branch of Chemistry and Chemical Engineering Guangdong Laboratory (HJL202202A009), the National Natural Science Foundation of China (Nos. 21207027 and 51372090), the Science and Technology Planning Project of Guangdong Province (No. 2017B090921002), the Scientific Research Project of the Department of Education of Guangdong Province (2022KQNCX046), the National Natural Science Foundation of China (32171708), and the Chaozhou Science and Technology Planning Project (Nos. 2018SS24 and 2021ZC29).

**Institutional Review Board Statement:** Not applicable.

**Informed Consent Statement:** Not applicable.

**Data Availability Statement:** Not applicable.

**Acknowledgments:** The innovative research team of advanced ceramic materials at Hanshan Normal University and the Guangdong Chaoshan Institute of Higher Education and Technology are both acknowledged.

**Conflicts of Interest:** The authors declare no conflict of interest. The funders had no role in the design of this study; in the collection, analyses, or interpretation of data; in the writing of the manuscript; or in the decision to publish the results.

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
