# Peer review of "Fabrication of Modified Polyurethane Sponge with Excellent Flame Retardant and the Modification Mechanism"

_coatings, doi:10.3390/coatings13040807_

Round 1

Reviewer 1 Report

The article needs a lot of corrections. This version is unpublishable.

There is no description of the type of foam that has been modified. The article does not have the required substantive value without a description of the material subjected to modification. Such studies cannot be verified.

1.      3.1. lowercase chapter title

2.      line 128 has no space before and

3.      line 134 wrong drawing number

4.      no caption under figure 4

5.      If we describe the flammability of foams, TGA should be tested in an oxygen atmosphere as described in the test methodology. At the same time, in the description of the results, line 148, it is written that foams were tested in a nitrogen atmosphere.

6.      line 164 Figure 4 has the wrong numbering 7.      In line 164, wrong capitalization in the description of the drawing 8.      Wrong figure and table number in row 168 9.      verse 180 chapter title in lower case 10.   Whether rinsing in ethanol did not change the state of the foam surface. Some polyurethanes degrade under the influence of alcohol 11.   There is no analysis of how much modifying material the foam has absorbed. There are foams that can absorb up to 300% of liquid. It would suffice to weigh the same sample before and after the modification 12.   Were no chlorine residues observed on the surface of the foams?

Author Response

1. The errors in labeling and chart numbers that appear in the text were revised.
2. Cleaning with alcohol is mainly for cleaning the impurities insoluble in water on the surface of polyurethane sponge, which will dry immediately after cleaning and will not cause degradation to it.
3. Because of the addition of inorganic flame retardants, in the quality will be increased, the experimental sample weight gain rate in 30%-50%.
4. Here is no chlorine residue. In the text is used in the inorganic flame retardant modification, the preparation process does not contain chlorine compounds.

Translated with www.DeepL.com/Translator (free version)

Reviewer 2 Report

This can be a nice work. The lack of cone calorimetry result is an obstacle to conclude on real flame behavior. I suggest authors to perform cone calorimetry and add data obtained. PCFC IS NOT SUFFICIENT AND RELEAVANT

Author Response

In this paper, we mainly combine microcalorimetry and thermogravimetry to analyze the flame retardant properties of the materials, whose samples are all in powder form and have certain contrast.

Round 2

Reviewer 1 Report

The authors still did not write what type of foams they subjected to modification by soaking. To say that it is a polyurethane sponge (PUS) is too general. There are many types of flexible polyurethane foams, which can be called ... polyurethane sponge ... Depending on the type of foam, they can absorb different amounts of liquid, i.e. in the modification process they can absorb different amounts of modifying materials. According to literature data, PUS can absorb from a few to hundreds of percent of the liquid. Without specifying what type of foam was modified, the experiment raises reservations as to its correctness. In response to the reviewer, the authors state that ... the experimental sample weight gain rate in 30%-50% ..... but do not state whether after the drying process after modification they checked the change in the weight of the sample or not. Based on the results of the TGA analysis presented by the authors, approx. 57% of the mass remained after the thermal degradation process, i.e. the analyzed sample contained approx. 50% of modifying agents. The amount of used modifying additives is very large. The method of analyzing the materials does not raise any objections. The subject matter of the article is very interesting, but the description of the materials before the modification and the foam after the modification is insufficient.

Author Response

Comment 1:

The authors still did not write what type of foams they subjected to modification by soaking. To say that it is a polyurethane sponge (PUS) is too general...The amount of used modifying additives is very large...but the description of the materials before the modification and the foam after the modification is insufficient.

Our response:

We thank the referee very much for the comments.

The polyurethane sponge with the density of 26 kg / m3 was purchased from Jin Tim Sponge Products Factory (Guangzhou China). The mass of additives is a little large, but the volume of the used modifying additives in the modified PUS is not large.

………………………………….end of response to the comments……………………………

Reviewer 2 Report

Authors claimed that a polyurethane sponge with excellent flame retardant was prepared, it should be proved by cone calorimeter test.

Author Response

Comment 1:

Authors claimed that a polyurethane sponge with excellent flame retardant was prepared, it should be proved by cone calorimeter test.

Our response:

We thank the referee very much for the comments.

The cone calorimeter test have been conducted by the FTT0007, the results suggested that the polyurethane sponge obtained excellent flame retardant after modification. The revised manuscript is as follows.

The flammabilities of the samples were evaluated using cone calorimetry. Some parameters used in this test include the smoke produce rate (SPR), the total smoke release (TSR), average CO yield (CO-Y), average CO2 yield (CO2-Y), time to ignition (TTI), ignition to flameout (ITF). Compared with the PUS, the modified PUS had longer TTI and ITF (Table 1) values because the modified PUS exhibited sustained burning with weak flames. A main function of flame retardants is to suppress the production of toxic smoke, providing time for people to evacuate the premises safely. The modification of PUS significantly reduced the TSR of PUS from 213 m2/m2 to 51 m2/m2 (Figure 3). Both peaks of CO-Y and CO2-Y decreased significantly after modification (Table 1). This implied that the modified surface served as an “insulating blanket” to intercept the exchangeable channel of heat flux and oxygen.

Table 1. Data from Cone Tests.

Samples

TTI (s)

 ITF (s)

CO-Y (kg/kg)

CO2-Y (kg/kg)

PUS

2

98

2.56

222.83

Modified PUS

29

154

0.16

1.51

 Figure 3. The the total smoke release (TSR) and the smoke produce rate (SPR) curves of PUS before and after modification.

………………………………….end of response to the comments……………………………
